# A FAST QUASI-NEWTON-TYPE METHOD FOR LARGE-SCALE STOCHASTIC OPTIMISATION

## ABSTRACT

During recent years there has been an increased interest in stochastic adaptations of limited memory quasi-Newton methods, which compared to pure gradient-based routines can improve the convergence by incorporating second order information. In this work we propose a direct least-squares approach conceptually similar to the limited memory quasi-Newton methods, but that computes the search direction in a slightly different way. This is achieved in a fast and numerically robust manner by maintaining a Cholesky factor of low dimension. This is combined with a stochastic line search relying upon fulfilment of the Wolfe condition in a backtracking manner, where the step length is adaptively modified with respect to the optimisation progress. We support our new algorithm by providing several theoretical results guaranteeing its performance. The performance is demonstrated on real-world benchmark problems which shows improved results in comparison with already established methods.

## 1 INTRODUCTION

In learning algorithms we often face the classical and hard problem of stochastic optimisation where we need to minimise some non-convex cost function $f(x)$

$$\min_{x \in \mathbb{R}^d} f(x), \tag{1}$$

when we only have access to *noisy* evaluations of the function and its gradients. We take a particular interest in situations where the number of data and/or the number of unknowns $d$ is very large.

The importance of this problem has been increasing for quite some time now. The reason is simple: many important applied problems ask for its solution, including most of the supervised machine learning algorithms when applied to large-scale settings. There are two important situations where the non-convex stochastic optimisation problem arise. Firstly, for large-scale problems it is often prohibitive to evaluate the cost function and its gradient on the entire dataset. Instead, it is divided into several mini-batches via subsampling, making the problem stochastic. This situation arise in most applications of deep learning. Secondly, when randomised algorithms are used to approximately compute the cost function and its gradients the result is always stochastic.

Our *contributions* are: 1. A new *stochastic line search* algorithm allowing for adaptive step-lengths akin to what is done in the corresponding state-of-the-art deterministic optimisation algorithms. This is enabled via a stochastic formulation of the first Wolfe condition. 2. We provide a new and efficient way of incorporating second-order (curvature) information into the stochastic optimiser via a *direct least-squares approach* conceptually similar to the popular limited memory quasi-Newton methods. 3. To facilitate a *fast and numerically robust implementation* we have derived tailored updating of a small dimension Cholesky factor given the new measurement pair (with dimension equal to the memory length). 4. We support our new developments by establishing several *theoretical properties* of the resulting algorithm. The performance is also demonstrated on *real-world benchmark problems* which shows improved convergence properties over current state-of-the-art methods.

## 2 RELATED WORK

Due to its importance, the stochastic optimisation problem is rather well studied by now. The first stochastic optimisation algorithm was introduced by Robbins & Monro (1951). It makes use of first-

order information only, motivating the name stochastic gradient (SG), which is the contemporary term (Bottou et al., 2018) for these algorithms, originally referred to as stochastic approximation. Interestingly most SG algorithms are not descent methods since the stochastic nature of the update can easily produce a new iterate corresponding to an increase in the cost function. Instead, they are Markov chain methods in that their update rule defines a Markov chain.

The basic first-order SG algorithms have recently been significantly improved by the introduction of various noise reduction techniques, see e.g. (Johnson & Zhang, 2013; Schmidt et al., 2013; Konečný & Richtárik, 2017; Defazio et al., 2014).

The well-known drawback of all first-order methods is the lack of curvature information. Analogously to the deterministic setting, there is a lot to be gained in extracting and using second-order information that is maintained in the form of the Hessian matrix. The standard quasi-Newton method is the BFGS method, named after its inventors (Broyden, 1967; Fletcher, 1970; Goldfarb, 1970; Shanno, 1970). In its basic form, this algorithm does not scale to the large-scale settings we are interested in. The idea of only making use of the most recent iterates and gradients in forming the inverse Hessian approximation was suggested by Nocedal (1980) and Liu & Nocedal (1989). The resulting L-BFGS method is computationally cheaper with a significantly reduced memory footprint. Due to its simplicity and good performance, this has become one of the most commonly used second-order methods for large-scale problems. Our developments makes use of the same trick underlying L-BFGS, but it is carefully tailored to the stochastic setting.

Over the past decade we have witnessed increasing capabilities of these so-called *stochastic quasi-Newton methods*, the category to which our developments belong. The work by Schraudolph et al. (2007) developed modifications of BFGS and its limited memory version. There has also been a series of papers approximating the inverse Hessian with a diagonal matrix, see e.g. Bordes et al. (2009) and Duchi et al. (2011). The idea of exploiting regularisation together with BFGS was successfully introduced by Mokhtari & Ribeiro (2014). Later Mokhtari & Ribeiro (2015) also developed a stochastic L-BFGS algorithm without regularisation. The idea of replacing the stochastic gradient difference in the BFGS update with a subsampled Hessian-vector product was recently introduced by Byrd et al. (2016), and Wang et al. (2017) derived a damped L-BFGS method.

Over the past five years we have also seen quite a lot of fruitful activity in combining the stochastic quasi-Newton algorithms with various first-order noise reduction methods (Moritz et al., 2016; Gower et al., 2016). A thorough and forward-looking overview of SG and its use within a modern machine learning context is provided by Bottou et al. (2018). It also includes interesting accounts of possible improvements along the lines of first-order noise reduction and second-order methods.

It is interesting—and perhaps somewhat surprising—to note that it is only very recently that stochastic line search algorithms have started to become available. One nice example is the approach proposed by Mahsereci & Hennig (2017) which uses the framework of Gaussian processes and Bayesian optimisation. The step length is chosen that best satisfies a probabilistic measure combining reduction in the cost function with satisfaction of the Wolfe conditions. Conceptually more similar to our procedure is the line search proposed by Bollapragada et al. (2018), which is tailored for problems that are using sampled mini-batches, as is common practice within deep learning.

## 3 STOCHASTIC LINE SEARCH

In deterministic line search algorithms we first compute a search direction $p_k$ and then decide how far to move along that direction according to

$$x_{k+1} = x_k + \alpha_k p_k, \tag{2}$$

where $\alpha_k > 0$ is referred to as the step length. The search direction is of the form

$$p_k = -H_k g_k, \tag{3}$$

where $H_k$ denotes an approximation of the inverse Hessian matrix. The question of how far to move in this direction can be framed as the following scalar minimisation problem

$$\min_{\alpha} f(x_k + \alpha p_k), \quad \alpha > 0. \tag{4}$$

The most common way of dealing with this problem is to settle for a possibly sup-optimal solution that guarantees at least a sufficient decrease. Such at solution can be obtained by selecting a step length $\alpha_k$ that satisfies the following inequality

$$f(x_k + \alpha p_k) \leq f(x_k) + c\alpha_k g_k^\mathsf{T} p_k, \tag{5}$$

where $c \in (0, 1)$. The above condition is known as the first Wolfe condition (Wolfe, 1969; 1971) or the Armijo condition (Armijo, 1966).

While deterministic line search algorithms have been well established for a long time, their stochastic counterparts are still to a large extent missing. Consider the case when the measurements of the function and its gradient are given by

$$\widehat{f}_k = f(x_k) + e_k, \qquad \widehat{g}_k = g_k + v_k, \tag{6}$$

where $g_k \triangleq \nabla f(x)|_{x=x_k}$, and $e_k \in \mathbb{R}$ and $v_k \in \mathbb{R}^{d \times 1}$ denote noise on the function and gradient evaluations, respectively. Furthermore we assume that

$$\mathrm{E}\left[e_k\right] = b, \qquad \mathrm{Cov}[e_k] = \sigma_f^2, \qquad \mathrm{E}\left[v_k\right] = 0, \qquad \mathrm{Cov}[v_k] = \sigma_g^2 I. \tag{7a}$$

The challenge is that since $\widehat{f}_k$ and $\widehat{g}_k$ are random variables it is not obvious how to select a step length $\alpha_k$ that—in some sense—guarantees a descent direction.

We explore the idea of requiring equation 5 to be fulfilled in expectation when the exact quantities are replaced with their stochastic counterparts,

$$\mathrm{E}\left[\widehat{f}(x_k + \alpha\widehat{p}_k) - \widehat{f}(x_k) - c\alpha_k\widehat{g}_k^\mathsf{T}\widehat{p}_k\right] \leq 0, \tag{8}$$

where $\widehat{p}_k = -H_k\widehat{g}_k$. This is certainly one way in which we can reason about the Wolfe condition in the stochastic setting we are interested in. Although satisfaction of equation 8 does not leave any guarantees when considering a single measurement, it still serves as an important property that could be exploited to provide robustness for the entire optimisation procedure. To motivate our proposed algorithm we hence start by establishing the following results.

**lemma 1 (Stochastic Wolfe condition 1)** *Assume that i) the gradient estimates are unbiased , ii) the cost function estimates are possibly biased , and iii) a descent direction is ensured in expectation* $\mathrm{E}\left[\widehat{p}_k^\mathsf{T}\widehat{g}_k\right] < 0$. *Then (for small $\alpha$:s)*

$$\mathrm{E}\left[\widehat{f}(x_k + \alpha\widehat{p}_k)\right] \leq \mathrm{E}\left[\widehat{f}(x_k) + c\alpha_k\widehat{g}_k^\mathsf{T}\widehat{p}_k\right], \quad where \quad 0 < c < \bar{c} = \frac{p_k^\mathsf{T} g_k}{p_k^\mathsf{T} g_k - \sigma_g^2 \mathrm{Tr}\left(H\right)}. \tag{9}$$

**Proof 1** *See Appendix A.1.*

**lemma 2** *There exists a step length $\alpha_k > 0$ such that* $\mathrm{E}\left[\widehat{f}(x_k + \alpha_k\widehat{p}_k) - \widehat{f}(x_k)\right] < 0$.

**Proof 2** *See Appendix A.2*

Relying upon these results we propose a line search based on the idea of repeatedly decreasing the step length $\alpha_k$ until the stochastic version of the first Wolfe condition is satisfied. Pseudo-code for this procedure is given in Algorithm 1. An input to this algorithm is the search direction $\widehat{p}_k$, which can be computed using any preferred method. The step length is initially set to be the minimum of the "natural" step length 1 and the iteration dependent value $\xi/k$. In this way the initial step length is kept at 1 until $k > \xi$, a point after which it is decreased at the rate $1/k$. Then we check whether the new point $x_k + \alpha\widehat{p}_k$ satisfies the stochastic version of the first Wolfe condition. If this is not the case, we decrease $\alpha_k$ with the scale factor $\rho$. This is repeated until the condition is met, unless we hit an upper bound $\max\{0, \tau - k\}$ on the number of backtracking iterations, where $\tau > k$ is a positive integer. With this restriction the decrease of the step length is limited, and when $k \geq \tau$ we use the initial step length no matter if the Wolfe condition is satisfied or not. The purpose of $\xi$ is to facilitate convergence by reducing the step length as the minima is approached. The motivation of $\tau$ comes from arguing that the backtracking loop does not contribute as much when the step length is small, and hence it is reasonable to provide a limit on its reduction.

---

**Algorithm 1** Stochastic backtracking line search

---

**Require:** Iteration index $k$, spatial point $x_k$, search direction $\widehat{p}_k$, scale factor $\rho \in (0,1)$, reduction
limit $\xi \geq 1$, backtracking limit $\tau > 0$.
 1: Set the initial step length $\alpha_k = \min\{1, \xi/k\}$
 2: Set $i = 1$
 3: **while** $\widehat{f}(x_k + \alpha \widehat{p}_k) > \widehat{f}(x_k) + c\alpha_k \widehat{g}_k^\mathsf{T} \widehat{p}_k$ **and** $i \leq \max\{0, \tau - k\}$ **do**
 4:     Reduce the step length $\alpha_k \leftarrow \rho \alpha_k$
 5:     Set $i \leftarrow i + 1$
 6: **end while**
 7: Update $x_{k+1} = x_k + \alpha_k \widehat{p}_k$

---

## 4  COMPUTING THE SEARCH DIRECTION

In this section we address the problem of computing a search direction based on having a limited
memory available for storing previous gradients and associated iterates. The approach we adopt
is similar to limited memory quasi-Newton methods, but here we employ a direct least-squares
estimate of the inverse Hessian matrix rather than more well-known methods such as damped L-
BFGS and L-SR1. In contrast to traditional quasi-Newton methods, this approach does not strictly
impose neither symmetry or fulfilment of the secant condition. It may appear peculiar to relax
these requirements. However, in this setting is not obvious that enforced symmetry necessarily
produces a better search direction. Furthermore, the secant condition relies upon the rather strong
approximation that the Hessian matrix is constant between two subsequent iterations. Treating the
condition less strictly might be helpful when that approximation is poor, perhaps especially in a
stochastic environment. We construct a limited-memory inverse Hessian approximation in Section
4.1 and show how to update this representation in Section 4.2. In Section 4.3 we discuss a particular
design choice associated with the update and Section 4.4 provides a means to ensure that a descent
direction is calculated. The complete algorithm combining the procedure presented in this section
with the line search routine in Algorithm 1 is given in Appendix C.

### 4.1  QUASI-NEWTON INVERSE HESSIAN APPROXIMATIONS

According to the secant condition (see e.g. Fletcher (1987)), the inverse Hessian matrix $H_k$ should
satisfy

$$H_k y_k = s_k, \tag{10}$$

where $y_k = g_k - g_{k-1}$ and $s_k = x_k - x_{k-1}$. Since there are generally more unknown values in $H_k$
than can be determined from $y_k$ and $s_k$ alone, quasi-Newton methods update $H_k$ from a previous
estimate $H_{k-1}$ by solving regularised problems of the type

$$H_k = \arg\min_H \quad \|H - H_{k-1}\|_{F,W}^2 \quad \text{s.t.} \quad H = H^\mathsf{T}, \quad H y_k = s_k, \tag{11}$$

where $\|X\|_{F,W}^2 = \|XW\|_F^2 = \operatorname{trace}(W^\mathsf{T} X^\mathsf{T} X W)$ and the choice of weighting matrix $W$ results
in different algorithms (see Hennig (2015) for an interesting perspective on this). Examples of the
most common quasi-Newton methods are given in Appendix B.

We employ a similar approach and determine $H_k$ as the solution to the following regularised least-
squares problem

$$H_k = \arg\min_H \|H Y_k - S_k\|_F^2 + \lambda \|H - \bar{H}_k\|_F^2, \tag{12}$$

where $Y_k$ and $S_k$ hold a limited number of past $\widehat{y}_k$'s and $s_k$'s according to

$$Y_k \triangleq [\widehat{y}_{k-m+1}, \dots, \widehat{y}_k], \qquad S_k \triangleq [s_{k-m+1}, \dots, s_k]. \tag{13}$$

Here, $m << d$ is the memory limit and $\widehat{y}_k = \widehat{g}_k - \widehat{g}_{k-1}$ is an estimate of $y_k$. The regulator matrix
$\bar{H}_k$ acts as a prior on $H$ and can be modified at each iteration $k$. The parameter $\lambda > 0$ is used to
control the relative cost of the two terms in equation 12. It can be verified that the solution to the
above least-squares problem (12) is given by

$$H_k = \left(\lambda \bar{H}_k + S_k Y_k^\mathsf{T}\right) \left(\lambda I + Y_k Y_k^\mathsf{T}\right)^{-1}, \tag{14}$$

where $I$ denotes the identity matrix. The above inverse Hessian estimate can be used to generate a search direction in the standard manner by scaling the negative gradient, that is

$$\widehat{p}_k = -H_k \widehat{g}_k. \tag{15}$$

However, for large-scale problems this is not practical since it involves the inverse of a large matrix. To ameliorate this difficulty, we adopt the standard approach by storing only a minimal (limited memory) representation of the inverse Hessian estimate $H_k$. To describe this, note that the dimensions of the matrices involved are

$$H_k \in \mathbb{R}^{d \times d}, \qquad Y_k \in \mathbb{R}^{d \times m}, \qquad S_k \in \mathbb{R}^{d \times m}. \tag{16}$$

We can employ the Sherman–Morrison–Woodbury formula to arrive at the following equivalent expression for $H_k$

$$H_k = \left( \bar{H}_k + \lambda^{-1} S_k Y_k^\mathsf{T} \right) \left[ I - Y_k \left( \lambda I + Y_k^\mathsf{T} Y_k \right)^{-1} Y_k^\mathsf{T} \right]. \tag{17}$$

Importantly, the matrix inverse $\left( \lambda I + Y_k^\mathsf{T} Y_k \right)^{-1}$ is now by construction a positive definite matrix of size $m \times m$. Therefore, we construct and maintain a Cholesky factor of $\lambda I + Y_k^\mathsf{T} Y_k$ since this leads to efficient solutions. In particular, if we express this matrix via a Cholesky decomposition

$$R_k^\mathsf{T} R_k = \lambda I + Y_k^\mathsf{T} Y_k, \tag{18}$$

where $R_k \in \mathbb{R}^{m \times m}$ is an upper triangular matrix, then the search direction $\widehat{p}_k = -H_k \widehat{g}_k$ can be computed via

$$\widehat{p}_k = -\bar{H}_k z_k - \lambda^{-1} S_k (Y_k^\mathsf{T} z_k), \quad z_k = \widehat{g}_k - Y_k w_k, \quad w_k = R_k^{-1} \left( R_k^{-\mathsf{T}} \left( Y_k^\mathsf{T} \widehat{g}_k \right) \right). \tag{19}$$

Note that the above expressions for $\widehat{p}_k$ involve first computing $w_k$, which itself involves a computationally efficient forward-backward substitution (recalling that $R_k$ is an $m \times m$ upper triangular matrix and the memory length $m$ is typically 10–50). Furthermore, $\bar{H}_k$ is typically diagonal so that $\bar{H}_k z_k$ is also efficient to compute. The remaining operations involve four matrix-vector products and two vector additions. Therefore, for problems where $d >> m$ then the matrix-vector products will dominate the computational cost.

Constructing $R_k$ can be achieved in several ways. The so-called normal-equation method constructs the (upper triangular) part of $\lambda I + Y_k^\mathsf{T} Y_k$ and then employs a Cholesky routine, which produces $R_k$ in $O(n\frac{m(m+1)}{2} + m^3/3)$ operations. Alternatively, we can compute $R_k$ by applying Givens rotations or Householder reflections to the matrix $M_k = \left[ \sqrt{\lambda} I \quad Y_k^\mathsf{T} \right]^\mathsf{T}$. This costs $O(2m^2((n+m) - m/3))$ operations, and is therefore more expensive, but typically offers better numerical accuracy (Golub & Van Loan, 2012).

## 4.2 Fast and robust inclusion of new measurements

In order to maximise the speed, we have developed a method for updating a Cholesky factor given the new measurement pair $(s_{k+1}, \widehat{y}_{k+1})$. Suppose we start with a Cholesky factor $R_k$ at iteration $k$ such that

$$R_k^\mathsf{T} R_k = \lambda I + Y_k^\mathsf{T} Y_k. \tag{20}$$

Assume, without loss of generality, that $Y_k$ and $S_k$ are ordered in the following manner

$$Y_k \triangleq [\mathcal{Y}_1, \widehat{y}_{k-m+1}, \mathcal{Y}_2], \qquad S_k \triangleq [\mathcal{S}_1, s_{k-m+1}, \mathcal{S}_2], \tag{21}$$

where $\mathcal{Y}_1, \mathcal{Y}_2, \mathcal{S}_1$ and $\mathcal{S}_2$ are defined as

$$\mathcal{Y}_1 \triangleq [\widehat{y}_{k-m+\ell+1}, \ldots, \widehat{y}_k], \qquad \mathcal{Y}_2 \triangleq [\widehat{y}_{k-m+2}, \ldots, \widehat{y}_{k-m+\ell}], \tag{22a}$$

$$\mathcal{S}_1 \triangleq [s_{k-m+\ell+1}, \ldots, s_k], \qquad \mathcal{S}_2 \triangleq [s_{k-m+2}, \ldots, s_{k-m+\ell}], \tag{22b}$$

and $\ell$ is an appropriate integer so that $Y_k$ and $S_k$ have $m$ columns. The above ordering arises from "wrapping-around" the index when storing the measurements. We create the new $Y_{k+1}$ and $S_{k+1}$ by replacing the oldest column entries, $\widehat{y}_{k-m+1}$ and $s_{k-m+1}$, with the latest measurements $\widehat{y}_{k+1}$ and $s_{k+1}$, respectively, so that

$$Y_{k+1} \triangleq [\mathcal{Y}_1, \widehat{y}_{k+1}, \mathcal{Y}_2], \qquad S_{k+1} \triangleq [\mathcal{S}_1, s_{k+1}, \mathcal{S}_2]. \tag{23}$$

The aim is to generate a new Cholesky factor $R_{k+1}$ such that

$$R_{k+1}^\mathsf{T} R_{k+1} = \lambda I + Y_{k+1}^\mathsf{T} Y_{k+1}. \tag{24}$$

To this end, let the upper triangular matrix $R_k$ be written conformally with the columns of $Y_k$ as

$$R_k = \begin{bmatrix} \mathcal{R}_1 & r_1 & \mathcal{R}_2 \\ \cdot & r_2 & r_3 \\ \cdot & \cdot & \mathcal{R}_4 \end{bmatrix}, \tag{25}$$

so that $\mathcal{R}_1$ and $\mathcal{R}_2$ have the same number of columns as $\mathcal{Y}_1$ and $\mathcal{Y}_2$, respectively. Furthermore, $r_1$ is a column vector, $r_2$ is a scalar and $r_3$ is a row vector. Therefore,

$$\begin{aligned}
R_k^\mathsf{T} R_k &= \begin{bmatrix} \mathcal{R}_1^\mathsf{T} \mathcal{R}_1 & \mathcal{R}_1^\mathsf{T} r_1 & \mathcal{R}_1^\mathsf{T} \mathcal{R}_2 \\ \cdot & r_2^2 + r_1^\mathsf{T} r_1 & r_1^\mathsf{T} \mathcal{R}_2 + r_2 r_3 \\ \cdot & \cdot & \mathcal{R}_4^\mathsf{T} \mathcal{R}_4 + \mathcal{R}_2^\mathsf{T} \mathcal{R}_2 + r_3^\mathsf{T} r_3 \end{bmatrix} \\
&= \begin{bmatrix} \lambda I + \mathcal{Y}_1^\mathsf{T} \mathcal{Y}_1 & \mathcal{Y}_1^\mathsf{T} \widehat{y}_{k-m+1} & \mathcal{Y}_1^\mathsf{T} \mathcal{Y}_2 \\ \cdot & \lambda + \widehat{y}_{k-m+1}^\mathsf{T} \widehat{y}_{k-m+1} & \widehat{y}_{k-m+1}^\mathsf{T} \mathcal{Y}_2 \\ \cdot & \cdot & \lambda I + \mathcal{Y}_2^\mathsf{T} \mathcal{Y}_2 \end{bmatrix}.
\end{aligned} \tag{26}$$

By observing a common structure for the update $\lambda I + Y_{k+1}^\mathsf{T} Y_{k+1}$ it is possible to write

$$\begin{aligned}
\lambda I + Y_{k+1}^\mathsf{T} Y_{k+1} &= \begin{bmatrix} \lambda I + \mathcal{Y}_1^\mathsf{T} \mathcal{Y}_1 & \mathcal{Y}_1^\mathsf{T} \widehat{y}_{k+1} & \mathcal{Y}_1^\mathsf{T} \mathcal{Y}_2 \\ \cdot & \lambda + \widehat{y}_{k+1}^\mathsf{T} \widehat{y}_{k-m+1} & \widehat{y}_{k+1}^\mathsf{T} \mathcal{Y}_2 \\ \cdot & \cdot & \lambda I + \mathcal{Y}_2^\mathsf{T} \mathcal{Y}_2 \end{bmatrix} \\
&= \begin{bmatrix} \mathcal{R}_1^\mathsf{T} \mathcal{R}_1 & \mathcal{R}_1^\mathsf{T} r_4 & \mathcal{R}_1^\mathsf{T} \mathcal{R}_2 \\ \cdot & r_5^2 + r_4^\mathsf{T} r_4 & r_4^\mathsf{T} \mathcal{R}_2 + r_5 r_6 \\ \cdot & \cdot & \mathcal{R}_6^\mathsf{T} \mathcal{R}_6 + \mathcal{R}_2^\mathsf{T} \mathcal{R}_2 + r_6^\mathsf{T} r_6 \end{bmatrix},
\end{aligned} \tag{27}$$

where $r_4$, $r_5$ and $r_6$ are determined by

$$r_4 = \mathcal{R}_1^{-\mathsf{T}}(\mathcal{Y}_1^\mathsf{T} \widehat{y}_{k+1}), \quad r_5 = \left(\lambda + \widehat{y}_{k+1}^\mathsf{T} \widehat{y}_{k+1} - r_4^\mathsf{T} r_4\right)^{1/2}, \quad r_6 = \frac{1}{r_5}\left(\widehat{y}_{k+1}^\mathsf{T} \mathcal{Y}_2 - r_4^\mathsf{T} \mathcal{R}_2\right). \tag{28}$$

The final term $\mathcal{R}_6$ can be obtained by noticing that

$$\mathcal{R}_6^\mathsf{T} \mathcal{R}_6 + \mathcal{R}_2^\mathsf{T} \mathcal{R}_2 + r_6^\mathsf{T} r_6 = \mathcal{R}_4^\mathsf{T} \mathcal{R}_4 + \mathcal{R}_2^\mathsf{T} \mathcal{R}_2 + r_3^\mathsf{T} r_3, \tag{29}$$

which implies

$$\mathcal{R}_6^\mathsf{T} \mathcal{R}_6 = \mathcal{R}_4^\mathsf{T} \mathcal{R}_4 - r_6^\mathsf{T} r_6 + r_3^\mathsf{T} r_3. \tag{30}$$

Therefore $\mathcal{R}_6$ can be obtained in a computationally very efficient manner by down-dating and updating the Cholesky factor $\mathcal{R}_4$ with the rank-1 matrices $r_6^\mathsf{T} r_6$ and $r_3^\mathsf{T} r_3$, respectively (see e.g. Section 12.5.3 in Golub & Van Loan (2012)).

## 4.3 SELECTING $\bar{H}_k$

There is no magic way of selecting the prior matrix $\bar{H}_k$. However, in practise it has proved very useful to employ a simple strategy of $\bar{H}_k \triangleq \gamma_k I$, where the positive scalar $\gamma_k > 0$ is adaptively chosen in each iteration. As a crude measure of progress we adopt the following rule

$$\gamma_k = \begin{cases} \kappa \gamma_{k-1}, & \text{if } \alpha_{k-1} = 1, \\ \gamma_{k-1}/\kappa, & \text{if } \alpha_{k-1} < 1/\rho^q, \\ \gamma_{k-1}, & \text{otherwise}, \end{cases} \tag{31}$$

where $\kappa \geq 1$ is a scale parameter, and $q$ corresponds to the number of backtracking loops in the line search; the values $\kappa = 1.3$ and $q = 3$ were found to work well in practise. The intuition behind equation 31 is that if no modification of the step length $\alpha_k$ is made, we can allow for a more "aggressive" regularisation. Note that a low $\gamma_k$ is favouring small elements in $H_k$. Since $\widehat{p}_k = -\bar{H}\widehat{g}_k$, this limits the magnitude of $\widehat{p}_k$ and the change $\|x_{k+1} - x_k\|$ is kept down. Hence, it is good practice to set the initial scaling $\gamma_0$ relatively small and then let it scale up as the optimisation progresses. Furthermore, we should point out that a diagonal $\bar{H}_k$ comes with an efficiency benefit, since the product $\bar{H}_k z_k$ in equation 19 then is obtained as the element-wise product between two vectors.

### 4.4 Ensuring a descent direction

In deterministic quasi-Newton methods, the search direction $p_k$ must be chosen to ensure a descent direction such that $p_k^\mathsf{T} g_k < 0$, since this guarantees reduction in the cost function for sufficiently small step lengths $\alpha_k$. Since $p_k = -Hg_k$, we have that $p_k^\mathsf{T} g_k = -g_k^\mathsf{T} H_k g_k$ which is always negative if the approximation $H_k$ of the inverse Hessian is positive definite. Otherwise, we can modify the search direction by subtracting a multiple of the gradient $p_k \leftarrow p_k - \beta_k g_k$. This is motivated by noticing that

$$(p_k - \beta g_k)^\mathsf{T} g_k = p_k^\mathsf{T} g_k - \beta_k g_k^\mathsf{T} g_k, \tag{32}$$

which always can be made negative by selecting $\beta_k$ large enough, i.e. if

$$\beta_k > \frac{p_k^\mathsf{T} g_k}{g_k^\mathsf{T} g_k}. \tag{33}$$

In the stochastic setting, the condition above does not strictly enforce a descent direction. Hence the search direction $\widehat{p}_k$ as determined by equation 15 is not a descent direction in general. However, ensuring that the condition is fulfilled in expectation is necessary since this is one of the assumptions made in lemma 1. Hence, we now establish the following result.

**lemma 3** *If*

$$\beta_k > \frac{p_k^\mathsf{T} g_k - \sigma_g^2 \operatorname{Tr}(H)}{g_k^\mathsf{T} g_k + d\sigma_g^2}, \tag{34}$$

*then*

$$\mathrm{E}\left[(\widehat{p}_k - \beta_k \widehat{g}_k)^\mathsf{T} \widehat{g}_k\right] < 0. \tag{35}$$

**Proof 3** *See Appendix A.3.*

In the practical setting we can not use equation 34 as it is, since we do not have access to $g_k$ and $p_k$, nor the noise variance $\sigma_g^2$. Instead we suggest a pragmatic approach in which these quantities are replaced by their estimates $\widehat{g}_k$, $\widehat{p}_k$ and $\widehat{\sigma_g^2}$. The noise estimate $\widehat{\sigma_g^2}$ could either be regarded a design parameter or empirically calculated from one or more sets of repeatedly collected gradient measurements. Nevertheless, picking $\beta_k$ sufficiently large ensures fulfilment of equation 32 and equation 35 simultaneously.

## 5 Numerical experiments

Let us now put our new developments to the test on a suite of problems from different categories to exhibit different properties and challenges. In Section 5.1 we study a commonly used benchmark, namely the collection of logistic classification problems described by Chang & Lin (2011) in the form of their library for support vector machines (LIBSVM). In Section 5.2 we consider an optimisation problem arising from the use of deep learning to solve the classical machine learning benchmark MNIST[1], where the task is to classify images of handwritten digits. Also, we test our method on training a neural network on the CIFAR-10 dataset (Krizhevsky, 2009).

In our experiments we compare against relevant state-of-the-art methods. All experiments were run on a MacBook Pro 2.8GHz laptop with 16GB of RAM using Matlab 2018b. All routines where programmed in C and compiled via Matlab's `mex` command and linked against Matlab's Level-1,2 BLAS libraries. More details about the experiments are available in Appendix D. The source code used to produce the results will be made freely available.

### 5.1 Logistic loss and a 2-norm regulariser

The task here is to solve eight different empirical risk minimisation problems using a logistic loss function with an L2 regulariser (two are shown here and all eight are profiled in Appendix D).

---

[1]`yann.lecun.com/exdb/mnist/`

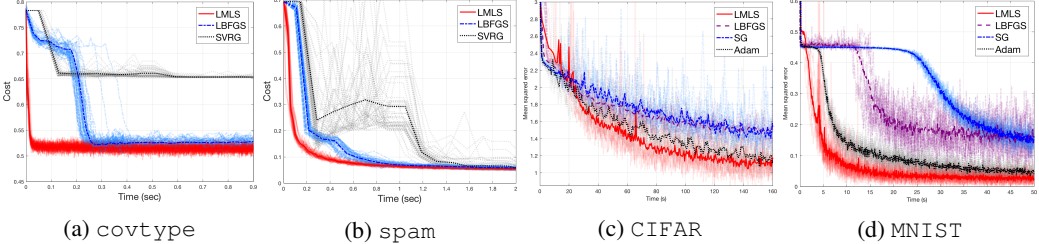

(a) `covtype`    (b) `spam`    (c) `CIFAR`    (d) `MNIST`

Figure 1: Performance on two classification tasks using a logistic loss with a two-norm regulariser ((a) and (b)), and two deep convolutional neural networks (CNNs) used for recognising images of handwritten digits from the MNIST data (c), and classification of the images in the CIFAR-10 data (d). Lighter shaded lines indicate individual runs, whereas the darker shaded line indicates the average.

The data is taken from Chang & Lin (2011). These problems are commonly used for profiling optimisation algorithms of the kind introduced in this paper, facilitating comparison with existing state-of-the-art algorithms. More specifically, we have used a similar set-up as Gower et al. (2016), which inspired this study. The chosen algorithm parameters for each case is detailed in Appendix D.

We compared our limited memory least-squares approach (denoted as `LMLS`) against two existing methods from the literature, namely, the limited memory stochastic BFGS method after Bollapragada et al. (2018) (denoted as `LBFGS`) and the stochastic variance reduced gradient descent (SVRG) by Johnson & Zhang (2013) (denoted `SVRG`). Figures 1a and 1b show the cost versus time for 50 Monte-Carlo experiments.

## 5.2 DEEP LEARNING

Deep convolutional neural networks (CNNs) with multiple layers of convolution, pooling and non-linear activation functions are delivering state-of-the-art results on many tasks in computer vision. We are here borrowing the stochastic optimisation problems arising in using such a deep CNN to solve the MNIST and CIFAR-10 benchmarks. The particular CNN structure used for the MNIST example employs $5 \times 5$ convolution kernels, pooling layers and a fully connected layer at the end. We made use of the publicly available code provided by Zhang (2016), which contains all the implementation details. For the CIFAR-10 example, the network includes 13-layers with more than 150,000 weights, see Appendix D for details. The MATLAB toolbox *MatConvNet* (Vedaldi & Lenc, 2015) was used in the implementation. Figures 1c and 1d show the average cost versus time for 10 Monte-Carlo trials with four different algorithms: 1. the method developed here (`LMLS`), 2. a stochastic limited memory BFGS method after Bollapragada et al. (2018) (denoted `LBFGS`), 3. `Adam` developed by Kingma & Ba (2015), and 4. stochastic gradient (denoted `SG`). Note that all algorithms make use of the same gradient code.

## 6 CONCLUSION AND FUTURE WORK

In this work we have presented a least-squares based limited memory optimisation routine that benefits from second order information by approximating the inverse Hessian matrix. The procedure is conceptually similar to quasi-Newton methods, although we do not explicitly enforce symmetry or satisfaction of the secant condition. By regularising with respect to an inverse Hessian prior, we allow for an adaptive aggressiveness in the search direction update. We have shown that the computations can be made robust and efficient using tailored Cholesky decompositions, with a cost that scales linearly in the problem dimension. Our method is designed for stochastic problems through a line search that repeatedly decreases the step length so as to satisfy the first Wolfe condition in expectation. Theoretical results have been established that support the proposed procedure. The method shows improved convergence properties over existing algorithms when applied to benchmark problems of various size and complexity.

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

# A   PROOFS OF THE LEMMAS

## A.1   THE FIRST STOCHASTIC WOLFE CONDITION

In the interest of a simple notation we note that we can drop the sub-index $k$ from all variables since all reasoning in this proof is for iteration $k$. We also drop $x$ from the arguments and write $f$ rather than $f(x)$, etc. However, we do make the randomness explicit, since this is crucial for arriving at the correct answer. For example we write $\widehat{f}_z$ to indicate that the random variable $z$ was used in computing the estimate of the function value at the iterate $x_k$. This means that the noise contaminating the measurement corresponds to the realisation of $z$.

Consider the stochastic version of the Wolfe condition

$$\mathrm{E}_{z,z'}\left[\widehat{f}_{z'}(x + \alpha\widehat{p}_z) - \widehat{f}_z(x) - c\alpha\widehat{p}_z^\mathsf{T}\widehat{g}_z\right] \le 0, \tag{36}$$

where the expected value is w.r.t. the randomness used in computing the required estimators as indicated by $z$ and $z'$. Using Taylor series we can for small step lengths $\alpha$ express $\widehat{f}_{z'}(x + \alpha\widehat{p}_z)$ according to

$$\widehat{f}_{z'}(x + \alpha\widehat{p}_z) \approx \widehat{f}_{z'} + \alpha\widehat{p}_z^\mathsf{T}\widehat{g}_{z'}. \tag{37}$$

Here it is crucial to note that the randomness in the above estimate stem from two different sources. The search direction $\widehat{p}_z$ enters in the argument of the Taylor expansion which is why it has to be computed before actually performing the Taylor expansion, implying that the randomness is due to $z$ and not $z'$ for the search direction. Based on equation 36 we have the following expression for the stochastic Wolfe condition

$$\mathrm{E}_{z,z'}\left[\widehat{f}_{z'} - \widehat{f}_z - \alpha\widehat{p}_z^\mathsf{T}\widehat{g}_{z'} + c\alpha\widehat{p}_z^\mathsf{T}\widehat{g}_z\right] \le 0. \tag{38}$$

Let $b$ denote a possible bias in the cost function estimator, then we have that $\mathrm{E}_{z'}\left[\widehat{f}_{z'}\right] = f + b$ and $\mathrm{E}_z\left[\widehat{f}_z\right] = f + b$. Hence, the first two terms in equation 38 cancel and we have

$$c \le \frac{\mathrm{E}_{z,z'}\left[\widehat{p}_z^\mathsf{T}\widehat{g}_{z'}\right]}{\mathrm{E}_{z,z'}\left[\widehat{p}_z^\mathsf{T}\widehat{g}_z\right]} = \frac{\mathrm{E}_z\left[\widehat{p}_z^\mathsf{T}\right]\mathrm{E}_{z'}\left[\widehat{g}_{z'}\right]}{\mathrm{E}_z\left[\widehat{p}_z^\mathsf{T}\widehat{g}_z\right]} = \frac{p^\mathsf{T}g}{\mathrm{E}_z\left[\widehat{p}_z^\mathsf{T}\widehat{g}_z\right]}. \tag{39}$$

The denominator can be written as

$$\mathrm{E}_z\left[\widehat{p}_z^\mathsf{T}\widehat{g}_z\right] = \mathrm{E}_z\left[-\widehat{g}_z^\mathsf{T}H\widehat{g}_z\right] = -\mathrm{E}_z\left[\mathrm{Tr}\left(H\widehat{g}_z\widehat{g}_z^\mathsf{T}\right)\right] = -\mathrm{Tr}\left(H\,\mathrm{E}_z\left[\widehat{g}_z\widehat{g}_z^\mathsf{T}\right]\right). \tag{40}$$

Using the definition of covariance we have that

$$\mathrm{E}_z\left[\widehat{g}_z\widehat{g}_z^\mathsf{T}\right] = \mathrm{E}_z\left[\widehat{g}_z\right]\mathrm{E}_z\left[\widehat{g}_z^\mathsf{T}\right] + \mathrm{Cov}\widehat{g} = gg^\mathsf{T} + \sigma_g^2 I, \tag{41}$$

where we in the last equality made use of the assumption that the gradients are unbiased and that $\mathrm{Cov}\widehat{g} = \sigma_g I$. Summarising we have that

$$\mathrm{E}_z\left[\widehat{p}_z^\mathsf{T}\widehat{g}_z\right] = -\mathrm{Tr}\left(Hgg^\mathsf{T}\right) - \sigma_g^2\,\mathrm{Tr}\left(H\right) = p^\mathsf{T}g - \sigma_g^2\,\mathrm{Tr}\left(H\right), \tag{42}$$

and hence we get

$$c \le \frac{p^\mathsf{T}g}{p^\mathsf{T}g - \sigma_g^2\,\mathrm{Tr}\left(H\right)}. \tag{43}$$

Recalling the assurance given in lemma 3 with the associated proof in Appendix A.3, we can always modify $p$ to guarantee that this bound is positive.

## A.2   REDUCTION IN COST FUNCTION

In this section we show that there exists an $\alpha > 0$ such that

$$\mathrm{E}_{z'}\left[\widehat{f}_{z'}(x + \alpha p)\right] < \mathrm{E}_z\left[\widehat{f}_z(x)\right]. \tag{44}$$

We do this by studying the difference

$$\mathrm{E}_{z'}\left[\widehat{f}_{z'}(x + \alpha p)\right] - \mathrm{E}_z\left[\widehat{f}_z(x)\right] = \mathrm{E}_{z,z'}\left[\widehat{f}_{z'}(x + \alpha p) - \widehat{f}_z(x)\right], \tag{45}$$

in which the first term can be expressed using a Taylor series according to

$$\widehat{f}_{z'}(x + \alpha \widehat{p}_z) = \widehat{f}_{z'} + \alpha \widehat{p}_z^\mathsf{T} \widehat{g}_{z'} + \mathcal{O}(\alpha \|\widehat{p}_z\|^2). \tag{46}$$

For small values of $\alpha$ we can discard the term $\mathcal{O}(\alpha\|\widehat{p}_z\|^2)$ from this expression. Hence we have that

$$\mathrm{E}_{z,z'}\left[\widehat{f}_{z'} + \alpha \widehat{p}_z^\mathsf{T} \widehat{g}_{z'} - \widehat{f}_z\right] = \alpha \mathrm{E}_{z,z'}\left[\widehat{p}_z^\mathsf{T} \widehat{g}_{z'}\right] = \alpha \mathrm{E}_z\left[\widehat{p}_z^\mathsf{T}\right]\mathrm{E}_{z'}\left[\widehat{g}_{z'}\right] = \alpha p^\mathsf{T} g. \tag{47}$$

Following from Section 4 we can ensure $p^\mathsf{T} g$ to be negative, and hence equation 44 holds as long as $\alpha$ is chosen small enough for the Taylor expansion to be a valid approximation.

### A.3 ENSURING A DESCENT DIRECTION

Note that

$$\mathrm{E}_z\left[(\widehat{p}_z - \beta \widehat{g}_z)^\mathsf{T} \widehat{g}_z\right] = \mathrm{E}_z\left[\widehat{p}_z^\mathsf{T} \widehat{g}_z\right] - \beta \mathrm{E}_z\left[\widehat{g}_z^\mathsf{T} \widehat{g}_z\right] = \mathrm{E}_z\left[-\widehat{g}_z^\mathsf{T} H \widehat{g}_z\right] - \beta \mathrm{E}_z\left[\widehat{g}_z^\mathsf{T} \widehat{g}_z\right]. \tag{48}$$

Using equation 42 we have

$$\mathrm{E}_z\left[-\widehat{g}_z^\mathsf{T} H \widehat{g}_z\right] = -\mathrm{Tr}\left(Hgg^\mathsf{T}\right) - \sigma_g^2 \mathrm{Tr}\left(H\right) = p^\mathsf{T} g - \sigma_g^2 \mathrm{Tr}\left(H\right), \tag{49}$$

and it directly follows from this that

$$\mathrm{E}_z\left[\widehat{g}_z^\mathsf{T} \widehat{g}_z\right] = \mathrm{E}_z\left[-\widehat{g}_z^\mathsf{T} I \widehat{g}_z\right] = g^\mathsf{T} g + d\sigma_g^2. \tag{50}$$

Hence

$$\mathrm{E}_z\left[(\widehat{p}_z - \beta \widehat{g}_z)^\mathsf{T} \widehat{g}_z\right] = p^\mathsf{T} g - \sigma_g^2 \mathrm{Tr}\left(H\right) - \beta\left(g^\mathsf{T} g + d\sigma_g^2\right), \tag{51}$$

and we see that

$$\mathrm{E}_z\left[(\widehat{p}_z - \beta \widehat{g}_z)^\mathsf{T} \widehat{g}_z\right] < 0 \Leftrightarrow \beta > \frac{p^\mathsf{T} g - \sigma_g^2 \mathrm{Tr}\left(H\right)}{g^\mathsf{T} g + d\sigma_g^2}. \tag{52}$$

## B QUASI-NEWTON METHODS

The historically most popular quasi-Newton method is given by the BFGS algorithm (Broyden, 1970; Fletcher, 1970; Goldfarb, 1970; Shanno, 1970)

$$H_{k+1} = \left(I - \frac{s_k y_k^\mathsf{T}}{y_k^\mathsf{T} s_k}\right) H_k \left(I - \frac{y_k s_k^\mathsf{T}}{y_k^\mathsf{T} s_k}\right) + \frac{s_k s_k^\mathsf{T}}{y_k^\mathsf{T} s_k}. \tag{53}$$

A closely related version is obtained if the optimisation in equation 11 is done with respect to the Hessian matrix rather than to its inverse. This results in the so-called DFP formula

$$H_{k+1} = H_k - \frac{H_k y_k y_k^\mathsf{T} H_k}{y_k^\mathsf{T} H_k y_k} + \frac{s_k s_k^\mathsf{T}}{y_k^\mathsf{T} s_k}. \tag{54}$$

Both of these are rank 2 updates that ensure $H_k$ to be positive definite, an attractive feature in the sense that it guarantees a descent direction. However, it may cause problems in regions where the true inverse Hessian is indefinite. Another well-known alternative is the symmetric rank 1 (SR1) method

$$H_k = H_{k-1} + \frac{(s_k - H_{k-1} y_k)(s_k - H_{k-1} y_k)^\mathsf{T}}{(s_k - H_{k-1} y_k)^\mathsf{T} y_k}. \tag{55}$$

Apart from BFGS and DFP above, this is a rank 1 update that in general does not preserve positive definiteness. For this reason it has received a particular interest within the so-called trust-region framework, where its ability of producing indefinite inverse Hessian approximations is being regarded as a major strength (Nocedal & Wright, 2006).

A main drawback of quasi-Newton methods is that they scale poorly to large problems, since the number of elements in $H_k$ equals the square of the input dimension $d$. This has resulted in developments of so-called limited memory algorithms, which rely upon the idea of forming the search direction $p_k$ directly without the need of storing the full matrix $H_k$. To that end a number $m << d$ of past differences $y$ and $s$ are being stored in memory. A notable member of this family is the L-BFGS algorithm (Nocedal, 1980), which has been widely within large-scale optimisation. During recent years, not at least due to the growing number of deep learning applications, there has been an increasing interest in adapting deterministic limited memory methods to the stochastic setting (Wang et al., 2017; Moritz et al., 2016; Gower et al., 2016; Schraudolph et al., 2007; Bordes et al., 2009; Mokhtari & Ribeiro, 2015; Byrd et al., 2016; Bollapragada et al., 2018).

## C  RESULTING ALGORITHM

We summarise our ideas from Section 3 and 4 in Algorithm 2. The if-statement on line 5 is included to provide a safe-guard against numerical instability.

## D  EXPERIMENT DETAILS

Details for the datasets used in Section 5 are listed in Table 1, including the parameter choices we made in our algorithm. Here, $b$ denotes the mini-batch size. The results of all logistic regression experiments are collected in Figure 2 (including those already shown in Figure 1), and the neural network results in Figures 1d and 1c are provided in a more readable size in Figure 3a and 3b. Detailed information of the network structure in the CIFAR experiment is provided in the printout shown in Figure 4.

The adaptive procedure of selecting the inverse Hessian prior $\bar{H}_k$ did not have much impact in most of the logistic regression examples, and thus it was kept constant $\bar{H}_k = \gamma_0 I$ except for in the covtype and URL problems. In the CIFAR and MNIST problems, however, the procedure was found to be of significant importance.

Table 1: List of the datasets used in the experiments, where $n$ denotes the size of the dataset (column 2) and $d$ denotes the number of variables in the optimisation problem (column 3). The remaining columns list our design parameters, including the mini-batch size $b$.

| Problem | $n$ | $d$ | $b$ | $m$ | $\lambda$ | $\xi$ | $\tau$ | $\gamma_0$ |
|---|---|---|---|---|---|---|---|---|
| gisette | 6 000 | 5 000 | 770 | 20 | $10^{-4}$ | 50 | 10 | 10 |
| covtype | 581 012 | 54 | 7 620 | 55 | $10^{-4}$ | 100 | 5 | 100 |
| HIGGS | 11 000 000 | 28 | 66 340 | 28 | $10^{-4}$ | 20 | 5 | 100 |
| SUSY | 3 548 466 | 18 | 5 000 | 18 | $10^{-4}$ | 50 | 10 | 100 |
| epsilon | 400 000 | 2 000 | 1 000 | 20 | $10^{-4}$ | 150 | 1 | 100 |
| rcv1 | 20 242 | 47 236 | 710 | 10 | $10^{-4}$ | 50 | 10 | 600 |
| URL | 2 396 130 | 3 231 961 | 1548 | 5 | $10^{-4}$ | 200 | 50 | 100 |
| spam | 82 970 | 823 470 | 2048 | 2 | $10^{-4}$ | 150 | 10 | 200 |
| MNIST | 60 000 | 3898 | 1000 | 50 | $8 \cdot 10^{-4}$ | 150 | 50 | 1 |
| CIFAR | 50 000 | 150 000 | 200 | 20 | $10^{-2}$ | 100 | 10 | 0.5 |

---

**Algorithm 2** Stochastic quasi-Newton with line search

---

**Require:** An initial estimate $x_1$, a maximum number of iterations $k_{\max}$, memory limit $m$, regularisation parameter $\lambda$, scale factor $\rho \in (0, 1)$, reduction limit $\xi \geq 1$, backtracking limit $\tau > 0$

1: Set $k = 1$
2: **while** $k < k_{\max}$ **do**
3:    Obtain a measurement of the cost function and its gradient

$$\widehat{f}_k = f(x_k) + e_k,$$
$$\widehat{g}_k = g_k + v_k.$$

4:    Set $\bar{H}_k = \gamma_k I$ where

$$\gamma_k = \begin{cases} \kappa\gamma_{k-1}, & \text{if } \alpha_{k-1} = 1, \\ \gamma_{k-1}/\kappa, & \text{if } \alpha_{k-1} < 1/\rho^q, \\ \gamma_{k-1}, & \text{otherwise.} \end{cases}$$

5:    **if** $\widehat{y}_k^\mathsf{T} s_k > \epsilon\|s_k\|_2^2$ **then**
6:      **if** $k > m$ **then**
7:         Form $Y_k$ and $S_k$ by replacing the oldest vector-pairs in $Y_{k-1}$ and $S_{k-1}$ with $\widehat{y}_k$ and $s_k$
8:      **else**
9:         Form $Y_k$ and $S_k$ by adding $\widehat{y}_k$ and $s_k$ to $Y_{k-1}$ and $S_{k-1}$
10:     **end if**
11:    **else**
12:       Set $Y_k = Y_{k-1}$ and $S_k = S_{k-1}$
13:    **end if**
14:    Select $\widehat{p}_k$ as

$$\widehat{p}_k = -\bar{H}_k z_k - \lambda^{-1} S_k(Y_k^\mathsf{T} z_k),$$
$$z_k = \widehat{g}_k - Y_k w_k,$$
$$w_k = R_k^{-1}\left(R_k^{-\mathsf{T}}\left(Y_k^\mathsf{T}\widehat{g}_k\right)\right),$$

with details provided in Section 4.2.
15:    Set $\widehat{p}_k \leftarrow \widehat{p}_k - \beta_k\widehat{g}_k$ with $\beta_k$ where

$$\beta_k > \frac{\widehat{p}_k^\mathsf{T}\widehat{g}_k - \widehat{\sigma_g^2}\,\mathrm{Tr}\,(H)}{\widehat{g}_k^\mathsf{T}\widehat{g}_k + d\widehat{\sigma_g^2}}.$$

16:    Set $\alpha_k = \min\{1, \xi/k\}$
17:    Set $i = 1$
18:    **while** $\widehat{f}(x_k + \alpha\widehat{p}_k) > \widehat{f}(x_k) + c\alpha_k\widehat{g}_k^\mathsf{T}\widehat{p}_k$ **and** $i \leq \max\{0, \tau - k\}$ **do**
19:       Reduce the step length $\alpha_k \leftarrow \rho\alpha_k$
20:       Set $i \leftarrow i + 1$
21:    **end while**
22:    Update $x_{k+1} = x_k + \alpha_k\widehat{p}_k$
23:    Set $k \leftarrow k + 1$
24: **end while**

---

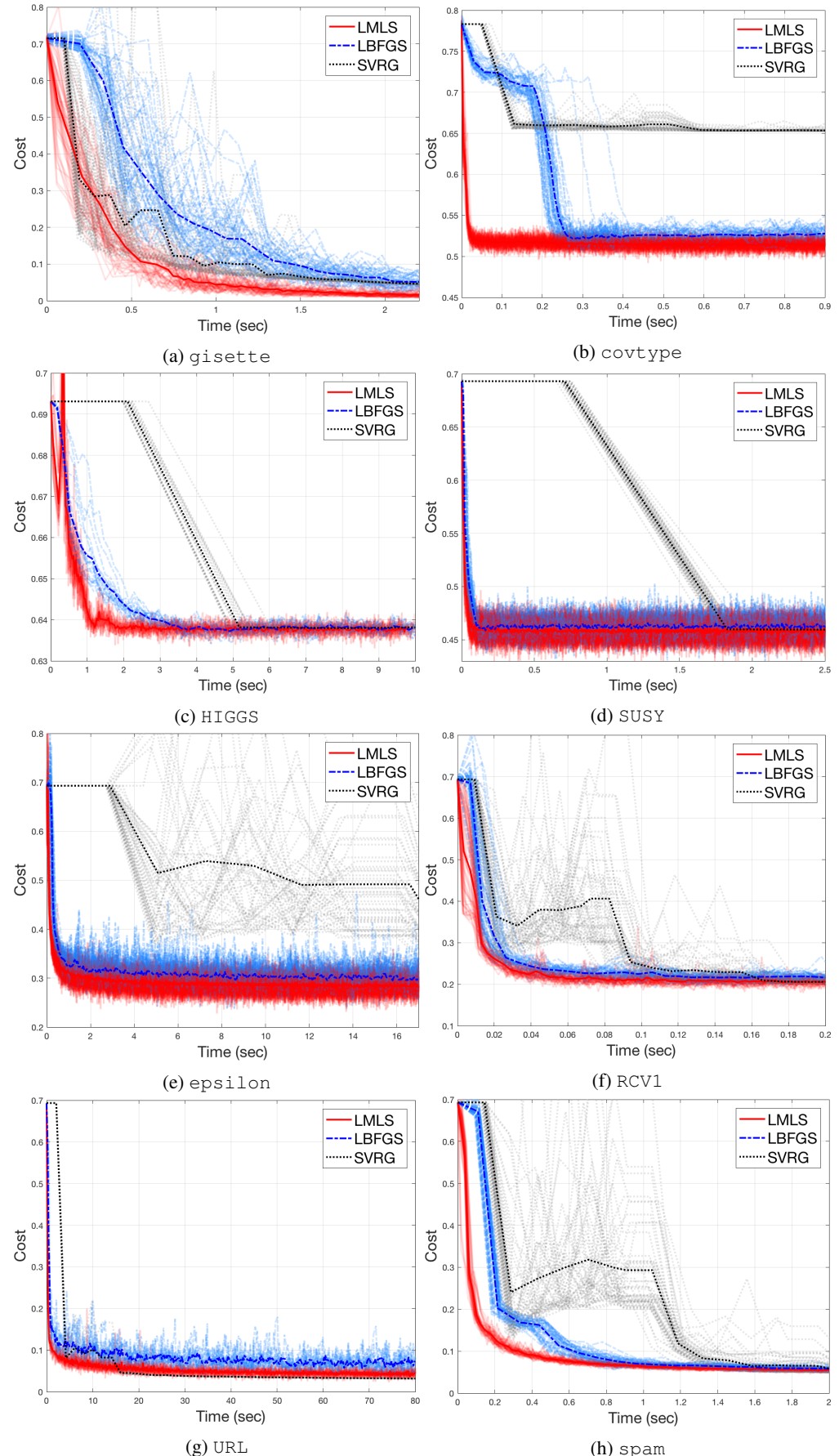

Figure 2: Performance on classification tasks using a logistic loss with a two-norm regulariser.

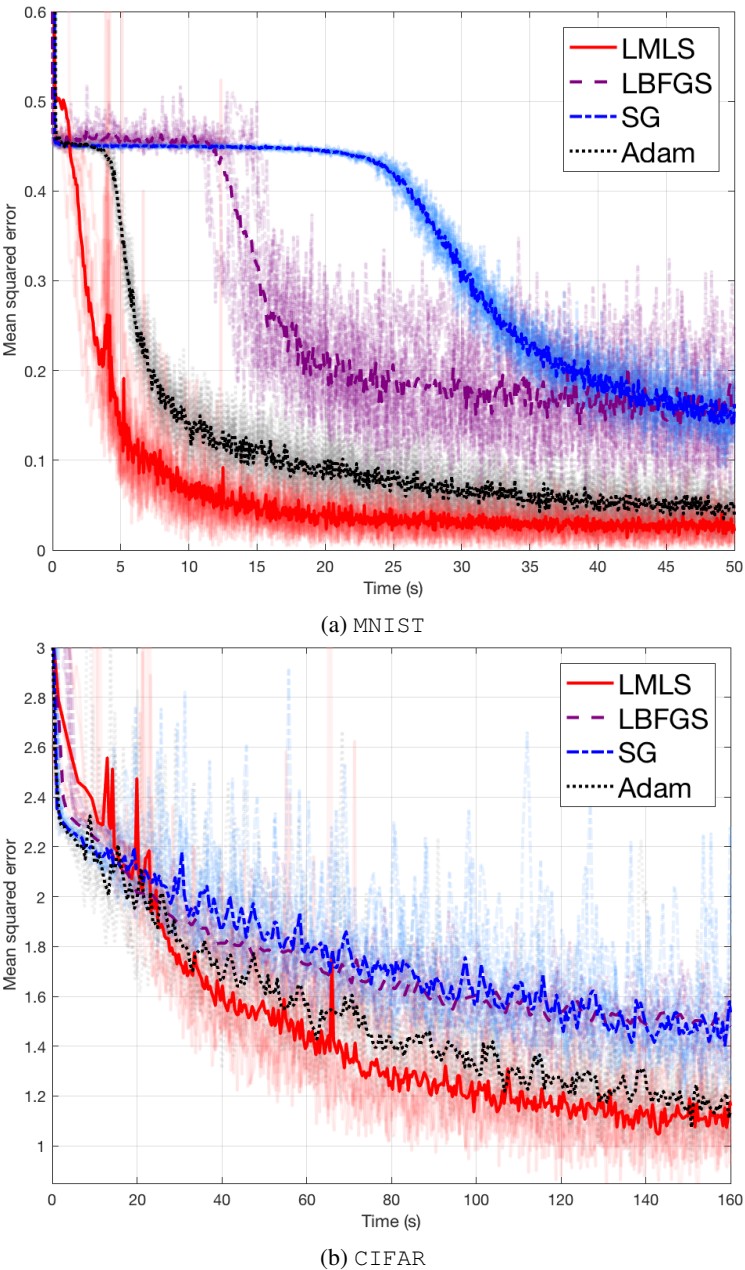

(a) MNIST

(b) CIFAR

Figure 3: Solving the optimisation problem used in training a state-of-the-art deep convolutional neural network (CNN) used for recognising images in the (a) MNIST and (b) CIFAR-10 dataset. LMLS refers to limited memory least-squares approach developed in this paper, SG refers to basic stochastic gradient and Adam refers to Kingma & Ba (2015).

```
   layer|     0|     1|      2|     3|     4|     5|      6|     7|     8|      9|     10|     11|     12|      13|
    type|input|  conv|  mpool|  relu|  conv|  relu|  apool|  conv|  relu|  apool|   conv|   relu|   conv|softmxl|
    name|  n/a|layer1| layer2|layer3|layer4|layer5| layer6|layer7|layer8| layer9|layer10|layer11|layer12|layer13|
----------|------|-------|------|------|------|-------|------|------|-------|-------|-------|-------|-------|
 support|  n/a|     5|      3|     1|     5|     1|      3|     5|     1|      3|      4|      1|      1|      1|
filt dim|  n/a|     3|    n/a|   n/a|    32|   n/a|    n/a|    32|   n/a|    n/a|     64|    n/a|     64|    n/a|
filt dilat|  n/a|    1|    n/a|   n/a|     1|   n/a|    n/a|     1|   n/a|    n/a|      1|    n/a|      1|    n/a|
num filts|  n/a|    32|    n/a|   n/a|    32|   n/a|    n/a|    64|   n/a|    n/a|     64|    n/a|     10|    n/a|
  stride|  n/a|     1|      2|     1|     1|     1|      2|     1|     1|      2|      1|      1|      1|      1|
     pad|  n/a|     2|0x1x0x1|     0|     2|     0|0x1x0x1|     2|     0|0x1x0x1|      0|      0|      0|      0|
----------|------|-------|------|------|------|-------|------|------|-------|-------|-------|-------|-------|
 rf size|  n/a|     5|      7|     7|    15|    15|     19|    35|    35|     43|     67|     67|     67|     67|
rf offset|  n/a|    1|      2|     2|     2|     2|      4|     4|     4|      8|     20|     20|     20|     20|
rf stride|  n/a|    1|      2|     2|     2|     2|      4|     4|     4|      8|      8|      8|      8|      8|
----------|------|-------|------|------|------|-------|------|------|-------|-------|-------|-------|-------|
data size|   32|    32|     16|    16|    16|    16|      8|     8|     8|      4|      1|      1|      1|      1|
data depth|    3|    32|     32|    32|    32|    32|     32|    64|    64|     64|     64|     64|     10|      1|
 data num|  200|   200|    200|   200|   200|   200|    200|   200|   200|    200|    200|    200|    200|      1|
----------|------|-------|------|------|------|-------|------|------|-------|-------|-------|-------|-------|
 data mem|  2MB|  25MB|    6MB|   6MB|   6MB|   6MB|    2MB|   3MB|   3MB|  800KB|   50KB|   50KB|    8KB|     4B|
param mem|  n/a|  10KB|     0B|    0B| 100KB|    0B|     0B| 200KB|    0B|     0B|  256KB|     0B|    3KB|     0B|

parameter memory|569KB (1.5e+05 parameters)|
    data memory| 61MB (for batch size 200)|
```

Figure 4: Detailed network strucure in the CIFAR-10 experiment.