# OpenReview forum: "A fast quasi-Newton-type method for large-scale stochastic optimisation"
_ICLR.cc/2019/Conference_

### Official Review · AnonReviewer1 · 2018-10-29
**A good attempt, but lacks sufficient explanation and reasoning**

**Rating:** 4
**Confidence:** 4

**Review:**

This paper presents a new quasi-Newton method for stochastic optimization that solves a regularized least-squares problem to approximate curvature information that relaxes both the symmetry and secant conditions typically ensured in quasi-Newton methods. In addition to this, the authors propose a stochastic Armijo backtracking line search to determine the steplength that utilizes an initial steplength of 1 but switches to a diminishing steplength in later iterations. In order to make this approach computationally tractable, the authors propose updating and maintaining a Cholesky decomposition of a crucial matrix in the Hessian approximation. Although it is a good attempt at developing a new method, the paper ultimately lacks a convincing explanation (both theoretical and empirical) supporting their ideas, as I will critique below.

1. Stochastic Line Search

Determining a steplength in the stochastic setting is a difficult problem, and I appreciate the authors’ attempt to attack this problem by looking at stochastic line searches. However, the paper lacks much detail and rigorous reasoning in the description and proof for the stochastic line search.

First, the theory gives conditions that the Armijo condition holds in expectation. Proving anything about stochastic line searches is particularly difficult, so I’m on board with proving a result in expectation and doing something different in practice. However, much of the detail on how this is implemented in practice is lacking.

How are the samples chosen for the line search? If we go along with the proposed theory, then when the function is reevaluated in the line search, a new sample is used. If this is the case, can one guarantee that the practical Armijo condition will hold? How often does the line search fail? How does the choice of the samples affect the cost of evaluating the line search?

The theory also suggests that the particular choice of c is dependent on each iteration, particularly the inner product between the true search direction and the true gradient at iteration k. Does this allow for a fixed c to be used? How is c chosen? Is it fixed or adaptive? What happens as the true gradient approaches 0?

The algorithm also places a limit on the number of backtracks permitted that decreases as the iteration count increases. What does the algorithm do when the line search fails? Does one simply take the step although the Armijo condition does not hold?

In deterministic optimization, BFGS typically needs a smaller steplength in the beginning as the algorithm learns the scale of the problem, then eventually accepts the unit steplength to obtain fast local convergence. The line search proposed here uses an initial steplength of $\min(1, \xi/k)$ so that in early iterations, a steplength of 1 is used and in later iterations the algorithm uses a $\xi/k$ steplength. When this is combined with the diminishing maximum number of backtracking iterations, this will eventually yield an algorithm with a steplength of $\xi/k$. Why is this preferred? Are the other algorithms in the numerical experiments tuned similarly?

The theory also asks for a descent direction to be ensured in expectation. However, it is not the case that $E[\hat{p}_k^T \hat{g}_k] = E[\hat{p}_k]^T g_k$, so it is not correct to claim that a descent direction is ensured in expectation. Rather, the condition is requiring the angle between the negative stochastic gradient direction and search direction to be acute in expectation.

All the proofs also depend on a linear Taylor approximation that is not well-explained, and I’m wary of proofs that utilize approximations in this way. Indeed, the precise statement is that $\hat{f}_{z’} (x + \alpha \hat{p}_z) = \hat{f}_{z’} + \alpha \hat{p}_z’ \hat{g}_z(x + \bar{\alpha} \hat{p}_z)$, where $\bar{\alpha} \in [0, \alpha]$. How does this affect the proof?

Lastly, I would propose for the authors to change the name of their condition to the “Armijo condition” rather than using the term “1st Wolfe condition” since the Wolfe condition is typically associated with the curvature condition (p_k’ g_new >= c_2 p_k’ g_k), hence referring to a very different line search.

2. Design of the Quasi-Newton Matrix

The authors develop an approach for designing the quasi-Newton matrix that does not strictly impose symmetry or the secant condition. The authors claim that this done because “it is not obvious that enforced symmetry necessarily produces a better search direction” and “treating the [secant] condition less strictly might be helpful when [the Hessian] approximation is poor”. This explanation seems insufficient to me to explain why relaxing these conditions via a regularized least-squares approach would yield a better algorithm, particularly in the noisy or stochastic setting. The lack of symmetry seems particularly strange; one would expect the true Hessian in the stochastic setting to still be symmetric, and one would still expect the secant condition to hold if the “true” gradients were accessible. It is also unclear how this approach takes advantage of the stochastic structure that exists within the problem.

Additionally, the quasi-Newton matrix is defined based on the solution of a regularized least squares problem with a regularization parameter lambda. It seems to me that the key to the approximation is the balance between the two terms in the objective. How is lambda chosen? What is the effect of lambda as a tuned parameter, and how does it affect the quality of the Hessian approximation? It is unclear to me how this could be chosen in a more systematic way.

The matrix also does not ensure positive definiteness, hence requiring a multiple of the gradient direction to be added to the search direction. In this case, the key parameter beta must be chosen carefully. What is a typical value of beta that is used for each of these problems? One would hope that beta is small, but if it is large, it may suggest that the search direction is primarily dominated by the stochastic gradient direction and hence the quasi-Newton matrix is not useful. The interplay of these different parameters needs to be investigated carefully.

Lastly, since (L-)BFGS use a weighted Frobenius norm, I am curious why the authors decided to use a non-weighted Frobenius norm to define the matrix. How does changing the norm affect the Hessian approximation?

All of these questions place the onus on the numerical experiments to see if these relaxations will ultimately yield a better algorithm.

3. Numerical Experiments

As written, although the range of problems is broad and the numerical experiments show much promise, I do not believe that I could replicate the experiments conducted in the paper. In particular, how is SVRG and L-BFGS tuned? How is the steplength chosen? What (initial) batch sizes are used? Is the progressive batching mechanism used? (If the progressive batching mechanism is not used, then the authors should refer to the original multi-batch paper by Berahas, et al. [1] which do not increase the batch size and use a constant steplength.)

In addition, a more fair comparison would include the stochastic quasi-Newton method in [2] that also utilize diminishing steplengths, which use Hessian-vector products in place of gradient differences. Multi-batch L-BFGS will only converge if the batch size is increased or the steplength diminished, and it’s not clear if either of these are done in the paper.

Typos/Grammatical Errors:
- Pg. 1: Commas are needed in some sentences, i.e. “Firstly, for large scale problems, it is…”; “…compute the cost function and its gradients, the result is…”
- Pg. 2: “Interestingly, most SG algorithms…”
- Pg 3: Remove “at least a” in second line
- Pg. 3: suboptimal, not sup-optimal
- Pg. 3: “Such a solution”, not “Such at solution”
- Pg. 3: Capitalize Lemma
- Pg. 4: fulfillment, not fulfilment
- Pg. 7: Capitalize Lemma
- Pg. 11: Before (42), Cov \hat{g} = \sigma_g^2 I
- Pg. 11: Capitalize Lemma

Summary:

In summary, although the ideas appear to provide better numerical performance, it is difficult to evaluate if the ideas proposed in this paper actually yield a better algorithm. Many algorithmic details are left unanswered, and the paper lacks mathematical or empirical evidence to support their claims. More experimental and theoretical work is needed before the manuscript can be considered for publication.

References:
[1] Berahas, Albert S., Jorge Nocedal, and Martin Takác. "A multi-batch l-bfgs method for machine learning." Advances in Neural Information Processing Systems. 2016.
[2] Byrd, Richard H., et al. "A stochastic quasi-Newton method for large-scale optimization." SIAM Journal on Optimization 26.2 (2016): 1008-1031.
[3] Schraudolph, Nicol N., Jin Yu, and Simon Günter. "A stochastic quasi-Newton method for online convex optimization." Artificial Intelligence and Statistics. 2007.

---

> ### Author Response · Authors · 2018-11-23
> **Response to Reviewer 1**
>
>
> Stochastic Line Search
> (1)
> It is correct that the function is repeatedly reevaluated in the line search, which is important for the method to work. This does not conflict with our claim that the Armijo condition holds in expectation and the associated proof. It is not entirely clear what ‘failure’ refers to, but we suppose it is in the sense of reaching the backtracking limit without having satisfied the Armijo condition; although we haven’t collected any statistics on how many times this situation occurred, it was clearly nothing that caused any problems. Regarding the cost, it is linearly proportional to the number of function evaluations as well as to the size of the batch.
>
> (2)
> The important insight is that there exists a value of c that, if chosen sufficiently small, ensures the Armijo condition to hold in expectation. In practise, we fixed c to 10^-6. This was found to work well and we did not see any need of modifying this value to improve the algorithm’s performance.
>
> (3)
> Yes, if the Armijo condition is not satisfied when the limit is reached, the step is still taken (this is akin to either a fixed or reducing learning rate in standard SGD methods). We motivate this by arguing that the backtracking loop does not contribute as much when the step length is small, and hence it is reasonable to a provide a limit on its reduction. The intention of the last paragraph in Section 3 is to clarify these details.
>
> (4)
> See comment (4) in the response to Reviewer 2.
>
> (5)
> We believe that there has been an misconception by the reviewer here. We do not make the claim $E[\hat{p}_k^T \hat{g}_k] = E[\hat{p}_k]^T g_k$ anywhere in the paper, and it is not a property that is being utilised. The claim we make is that a descent direction in expectation can be ensured by adding a multiple of the negative gradient estimate to the search direction estimate. That ensures that the scalar product between the stochastic gradient and the search direction is negative in expectation; this is equivalent to the angle between the negative stochastic gradient direction and search direction to be acute in expectation.
>
> (6)
> Once again, the reviewer is referring to a statement that is not in the paper, and therefore we can not respond to the comment.
>
> (7)
> Thank you for this suggestion, we agree that this would remove the risk of confusion.
>
> Design of Quasi-Newton Matrix
> (8)
> Regarding symmetry, see comment (1) in the response to Reviewer 2.
>
> The secant condition is an approximation that treats the Hessian approximation as constant between two subsequent iterations. In contrast to the claim, we can not expect this condition to hold even in the deterministic case, and hence it enforces a property that is certainly not valid for the true Hessian.
>
> (9)
> See comment (2) in the response to Reviewer 2.
>
> (10)
> In the implementation we used a pragmatic approach mimicking the deterministic scenario, and added 2 * p^T*g / g^T*g whenever p^T*g was non-negative. It should be mentioned that this situation occurred very infrequently (less than 10 times in total across all Monte Carlo experiments) and hence we saw no need to further investigate this particular parameter choice.
>
> Numerical Experiments
> (11)
> It is our clear intention to make the source code freely available to provide transparency and facilitate replication of the experiments. We have made a sincere effort in maximising the performance of the compared methods. Along with other parameter values, the batch sizes chosen are tabled in the Appendix. Regarding progressive batching, we used the method from Bollapragada 2018, but with a fixed batch size. The reason for this is that many other BFGS algorithms regularly failed to perform well, and in the interests of providing as fair a comparison as possible, we settled on the L-BFGS algorithm in Bollapragada 2018 with a fixed batch size.
>
> (12)
> We find it interesting that the machine learning community does not appear to have agreed on a benchmark quasi-Newton method that is effective across a wide range of problems. Indeed, the method in [2] and in many others make comparisons against a subset of alternative methods, but not across all possible quasi-Newton methods. Further, there is a sincere lack of software for second-order methods within the community. Therefore, in light of the rather strong comments from the reviewer in this regard, we are uncertain if any comparison will ever satisfy all reviewers. Our solution to this in the current manuscript was to implement what we considered to be some state-of-the-art methods against the proposed method.

---

> > ### Comment · AnonReviewer1 · 2018-11-24
> > **Quick clarification on comments/questions**
> >
> > Thanks to the authors for their response and answering some of my questions. I want to clarify some of my questions/points that I think were potentially unclear before.
> >
> > (1) Although we are interested in minimizing a function with errors, in practice, I assume that you are working with a sampled function. Are you using different samples when evaluating the Armijo condition? Or are you enforcing some kind of consistency between the sampled functions you are evaluating?
> >
> > I ask this because a few potential issues may arise if the sampled functions are inconsistent, namely: (1) the search direction may not be a descent direction on the sampled function, hence it is possible for the line search to fail in practice; and (2) depending on the amount of error, it is possible for the comparison of function values to be dominated by noise. If you’re using inconsistent function values, can you clarify how you address these two potential algorithmic issues?
> >
> > Just to be clear: line search failure here means when one has reached the maximum number of line search steps and has not satisfied the Armijo condition, as you’ve mentioned.
> >
> > (2) Thanks for clarifying the practical choice of c. The choice of c = 1e-6 should be mentioned in the manuscript. My original question, however, was based on the proposed theory. In particular, as $\|g_k\|$ goes to 0, is it possible for $\bar{c}$ to go to 0 as well or can one guarantee that this will always be bounded below for all $k$ (either theoretically or via some additional algorithmic mechanism)?
> >
> > (3-4) Thank you for the clarification on the workings of the line search. In light of this, if it’s possible for the Armijo condition to fail and yet the steplength is still used, then does the line search here still guarantee sufficient decrease on any sampled function at all? Or is it simply being used to diminish your steplength by a certain amount in bad cases in the initial O(1) phase?
> >
> > (5-6) Thanks for catching my mistake! I must have misinterpreted something in the proof or theory at the time.
> >
> > (11-12) My initial question was actually regarding the tuning of the compared algorithms (L-BFGS and SVRG). It would be good if the best value or range of parameters considered is given for these other algorithms. For SVRG, this would include the steplength and snapshot length.
> >
> > Regarding the comparison to Bollapragada, et al., the increasing batch size method was used in their method to reduce the variance so that an O(1) steplength may always be utilized (where the steplength is determined based on variance estimation and a stochastic line search). Can you clarify (including in the manuscript) which mechanisms you are using from their work (for instance, the line search, variance estimation, etc.)? In the comparisons, how is the variance reduced or controlled in their L-BFGS algorithm when using a fixed batch size in order to produce a fair comparison?
> >
> > (As noted above by the authors, the proposed algorithm utilizes an O(1/k) steplength asymptotically, which will control the variance eventually and yield convergence. It is not clear if the L-BFGS algorithm compared against here has any similar mechanism without the use of increasing batch sizes. This is why I had suggested comparing against [2], as it also utilizes an O(1/k) steplength, but an answer to the above for Bollapragada, et al.'s algorithm would also be sufficient.)
> >
> > I am open to improving the rating of this paper if these comments are well-addressed. Thanks again in advance!

---

> > > ### Author Response · Authors · 2018-11-26
> > > **Response to clarifying questions**
> > >
> > > (1) Response: We are indeed using different samples without any consistency enforcement, so any pair of function evaluations can be considered independent. It is correct that the line search may fail in the sense you have in view, and frequent failure would certainly slow down the progress of the algorithm. However, we didn’t find this to be an issue in practice, and hence we have not addressed it further. Importantly, when the steplength is small, then we are effectively comparing two random numbers (the function evaluations) that are independently chosen and that have very similar mean values (since the function arguments are almost identical), which implies - with high likelihood - that we will satisfy the Armijo-type condition.
> > >
> > > Regarding the second point, it is true that the optimisation becomes harder when the noise level is large in relation to the function values, which is a natural challenge in the stochastic setting. Although demonstrated on Deep Learning problems, our algorithm is not specifically designed for that field but rather for stochastic functions in general. Therefore we have not considered any customised sampling techniques.
> > >
> > > (2) Response: Thank you for pointing this out, we should include this clarification in the paper. You are absolutely right here: in the setting we present, $\bar{c}$ equals 0 whenever $\|g_k\|$ does, e.g. at local minimas. We have not investigated any guarantees of the kind you suggest. Instead we note that the choice of c was not found to cause problems in practice.
> > >
> > > (3-4) Response: What the theory does guarantee is that there is a non-zero step length such that a decrease is ensured in expectation. This is indeed not a strong result. However, it is not obvious how to strengthen the result without adding unrealistic assumptions. At the same time it is correct that the line search is designed to diminish the step length if the Armijo condition keeps failing; at most, the reduction continues down to machine epsilon precision. At the same time, when the steplength is small, then we are effectively comparing two random numbers that are independently chosen and that have very similar mean values, which implies - with high likelihood - that we will satisfy the Armijo-type condition before the steplength reaches machine epsilon.
> > >
> > > (11-12) Response: Thank you for this fair point, we are happy to make this clarification. We are using the mechanisms of Bollapragada that concerns the computation of the search direction, while for the remaining parts the proposed method is being employed. That is, when comparing the LMLS approach to L-BFGS, we are only employing the search direction part of Bollapragada while the line search procedure is common to both (including the ultimate O(1/k) steplength choice). Importantly, the L-BFGS approach in their paper requires that there is consistency between consecutive samples when forming the search direction and we have ensured that this is satisfied for the L-BFGS approach. In fact, it was discovered that this is essential for the method in Bollapragada to work in practice, while it is not required for the proposed LMLS approach.
> > >
> > > In terms of SVRG, we chose the snapshot length to equal the batch size for LMLS and L-BFGS approaches. We also chose the steplength according to the following Matlab code (essentially this is equal to 4.0 / (max_i || X(:,i) ||^2) )
> > >
> > > OPT.stepSize     = sum(p.X(:,1).^2);
> > > for i=2:size(p.X,2)
> > >     tmp = sum(p.X(:,i).^2);
> > >     if tmp > OPT.stepSize
> > >         OPT.stepSize = tmp;
> > >     end
> > > end
> > > OPT.stepSize = 1/(0.25*OPT.stepSize);
> > >
> > > These choices provided the most consistent performance across all considered problems in the paper.

---

### Official Review · AnonReviewer2 · 2018-10-31
**Novel idea, but not without issues to be addressed**

**Rating:** 5
**Confidence:** 5

**Review:**

This paper presents a new quasi-Newton type method for stochastic optimization problems. The primary contributions of the paper include a new stochastic linesearch method as well as a novel way to incorporate second order information which is different from existing approaches such as BFGS or L-BFGS.

In terms of the clarity, I think this is a very well-written paper with nice organization. The paper does have some typos, though.

In terms of significance, how to incorporate second-order information in stochastic optimization has long been an important research topic. Most existing stochastic quasi-Newton methods use L-BFGS method to incorporate second order information and choose a fixed, small stepsize, with the only differences being how to compute the curvature pair (s_k, y_k). Therefore, this paper is addressing a very important question and has made respectable attempt to use mechanisms other than L-BFGS method and to incorporate a linesearch scheme.

Specifically, the paper relaxes the secant equation, which is natural for the stochastic settings because the difference in gradients y_k is computed from stochastic gradients, and the true Hessian only satisfies the secant equation in expectation. I believe replacing the secant equation is an important and promising direction.

However, there are concerns about the new approach proposed in this paper:

1.	The resulting Hessian inverse approximation in (14) is no longer symmetric, or guaranteed to be positive definite. While the underling true Hessian might not be positive definite because of the nonconvexity, it is always symmetric. Is it possible to impose symmetricity as a constraint in (12)?

2.	What is the correct way to choose regularization parameter λ in (14)?

The paper also proposes a stochastic linesearch algorithm. For this part, there are several concerns as well:

1.	The assumption that the covariance of gradient estimator is a constant multiple of identity is a strong and unrealistic assumption, which is never satisfied in machine learning.

2.	The algorithm performs a backtracking linesearch, with the initial trial stepsize decreaing as O(1/k), which means that the stepsize used is always decreasing at least as fast as O(1/k). This is in general in stark contrast with the intuition that a O(1) stepsize should be used for a quasi-Newton method.

3.	Satisfying the Armijo condition in expectation does not lead to any useful convergence guarantee.

The paper also presented some numerical experiments. While the numerical results look promising, I would appreciate some clarification about what method they are really comparing against. For example, for LBFGS the authors cite R. Bollapragada et al. “A progressive batching L-BFGS method for machine learning”. Is the paper comparing against progressive batching L-BFGS? The results of LBFGS here seem to be very different from the paper cited.

Finally, the paper could certainly benefit by making some mathematical statement more rigorous. For example, Lemma 1 and 2 are stated in expectation; however, since the algorithm is a stochastic algorithm, the whole sequence {x_k} generated is a stochastic process, and the expectation in the lemmas are conditional expectations. It is important to clarify w.r.t. what the conditional expectation is taken.

In summary, I believe that this paper has made a novel contribution. However, the author should address the concerns above.

---

> ### Author Response · Authors · 2018-11-23
> **Response to Reviewer 2**
>
>
> (1)
> --Reviewer Comment: "The resulting Hessian inverse approximation in (14) is no longer symmetric, or guaranteed to be positive definite. While the underling true Hessian might not be positive definite because of the nonconvexity, it is always symmetric. Is it possible to impose symmetricity as a constraint in (12)?"
>
> Response: In terms of symmetry, while we understand that the true Hessian is symmetric by definition, it is not clear what benefit (if any) is gained by enforcing this constraint on a Hessian approximation. Indeed, the search direction is obtained by computing p = -Hg, and it is not obvious that symmetry is required to ensure that p is "better". At the same time, we also appreciate the long and successful history that quasi-Newton methods enjoy, and therefore we were reticent to make strong statements about symmetry.
>
> (2)
> --Reviewer Comment: "What is the correct way to choose regularization parameter λ in (14)?"
>
> Response: This is a great question that is not addressed in the paper. In fact, in the paper we have treated this as a "tuning" parameter and left choosing its optimal value as future work. How this can be done in a principled manner is not obvious, and similar tuning parameters are present in most optimisation algorithms. On a higher ground, this relates to the question of how to design an (appropriate) prior distribution in Bayesian methods, to which no precise answer exists.
>
> (3)
> --Reviewer Comment: "The assumption that the covariance of gradient estimator is a constant multiple of identity is a strong and unrealistic assumption, which is never satisfied in machine learning."
>
> Response: We completely agree with this comment. In fact, the assumption was made without further consideration, and the proofs would not suffer from a generalisation. This will certainly be updated in a revised version of the paper.
>
> (4)
> --Reviewer Comment: "The algorithm performs a backtracking linesearch, with the initial trial stepsize decreaing as O(1/k), which means that the stepsize used is always decreasing at least as fast as O(1/k). This is in general in stark contrast with the intuition that a O(1) stepsize should be used for a quasi-Newton method. "
>
> Response: The algorithm only starts to reduce the step-length after $\xi$ iterations (so the maximum step-length is actually chosen as min(1, \xi / k) ). This means that in the initial phases the step-length could be O(1), but may also be quite small if required to satisfy the line-search condition. On the other hand, after $\xi$ iterations, the step-size begins to reduce at O(1/k), which is primarily to ensure an eventual point-estimate.
>
> (5)
> --Reviewer Comment: "Satisfying the Armijo condition in expectation does not lead to any useful convergence guarantee"
>
> Response: We agree that this is not a strong result. At the same time, it is not immediately obvious how to strengthen it without adding unrealistic assumptions.
>
> (6)
> --Reviewer Comment: "The paper also presented some numerical experiments. While the numerical results look promising, I would appreciate some clarification about what method they are really comparing against. For example, for LBFGS the authors cite R. Bollapragada et al. “A progressive batching L-BFGS method for machine learning”. Is the paper comparing against progressive batching L-BFGS? The results of LBFGS here seem to be very different from the paper cited."
>
> Response: This is a very good observation. In fact, we are employing the L-BFGS method from R. Bollapragada et al. , but with fixed batch size (the same size used for the proposed method) so that the timing comparison is fair. Out of the half-dozen or so BFGS algorithms that were coded and trialled, this gave the most consistent performance. All other BFGS methods failed on certain problems.
>
> (7)
> --Reviewer Comment: "Finally, the paper could certainly benefit by making some mathematical statement more rigorous. For example, Lemma 1 and 2 are stated in expectation; however, since the algorithm is a stochastic algorithm, the whole sequence {x_k} generated is a stochastic process, and the expectation in the lemmas are conditional expectations. It is important to clarify w.r.t. what the conditional expectation is taken."
>
> Response: Thank you for this comment. We agree that some statements could be made clearer in the manuscript. However, we have taken inspiration from Bottou et al [1] where convergence is proven in expectation and the Markov property (which also applies here) is relied on.
>
> Reference
> [1] L. Bottou, F. E. Curtis, and J. Nocedal. Optimization methods for large-scale machine learning. SIAM Review, 60(2):223-311, 2018.

---

### Official Review · AnonReviewer3 · 2018-11-03
**interesting reformulation of Quasi Newton direction**

**Rating:** 5
**Confidence:** 5

**Review:**

The authors present an interesting variation of the standard QN methods. Their main point of departure from LBFGS/SR1 is in constructing a simpler Hessian inverse approximation. Recall that SR1 and LBFGS updates all satisfy the secant equation for each of the `m` previous gradient differences stored in memory. The authors choose to get "close" to satisfying the equations by solving an l_2 penalization of the secant equations.

The resulting algorithm is interesting, but it is not clear from the paper what the claimed advantage of doing this is. The LBFGS and SR1 unrolled update rules for H (Hessian inverse approximation) is O(m^2 d) (Sec 7.2 NW 2006),  and this seems to be the same for the authors' method, where the main matrix R_k that forms H has the same order. (BTW, did you mean 'd' in place of 'n'  the computational order discussion preceding Sec  4.2?)

The experiments show that this method's performance is impressive compared to an LBFGS implementation provided by Bollapragada 2018 , but as I recall that paper presented a variable/increasing batch method, while the authors' method uses fixed batches (as far as I can tell) so it is not clear that comparison on time alone is sufficient. The advantage over LBFGS and SGD seen in MNIST seems to go away by the CIFAR example, so it is unclear what might happen in larger problems like ImageNet.

I am also not able to see the difference between the 'stochastic' line search presented here and the standard backtracking method as applied to mini-batch evaluated estimates. What is different, and new that brings in consideration for the noise? I recall that bollapragada 2018 had an additional variance based rule to check. Some more conservative values are chosen for the step length, but I do not see the justification presented in the appendix, esp Eq 47 : p is not independent from g here, being calucated as p=Hg,  so E[p^t g] is not equal to the product of  the individual expectations.

Some key points were left out in the discussion of the experiments. This is a common slip up when writing conf papers these days, but please do consider discussing the settings of parameters like mini-batches sizes , value of \lambda in the H derivation, how one calculates the \sigma^2_g within the algorithm presented in the Appendix. The last must include
an extra computational cost, or are you using Adam style online variance estimator?

The MSE error alone seems insufficient in the results. Please publish the test mis-classification results too. Also, why is the MSE loss used with the softmax in CIFAR? Shouldn't cross-entropy, better justified theoretically, be better justified?

---

> ### Author Response · Authors · 2018-11-23
> **Response to Reviewer 3**
>
>
> (1)
> --Reviewer Comment: "The resulting algorithm is interesting, but it is not clear from the paper what the claimed advantage of doing this is. The LBFGS and SR1 unrolled update rules for H (Hessian inverse approximation) is O(m^2 d) (Sec 7.2 NW 2006),  and this seems to be the same for the authors' method, where the main matrix R_k that forms H has the same order. (BTW, did you mean 'd' in place of 'n' the computational order discussion preceding Sec 4.2?)"
>
> Response: Yes, we meant "d" and thank you for spotting this typo. You are quite correct that the computational load is similar between these two approaches.
>
> (2)
> --Reviewer Comment: "The experiments show that this method's performance is impressive compared to an LBFGS implementation provided by Bollapragada 2018 , but as I recall that paper presented a variable/increasing batch method, while the authors' method uses fixed batches (as far as I can tell) so it is not clear that comparison on time alone is sufficient. The advantage over LBFGS and SGD seen in MNIST seems to go away by the CIFAR example, so it is unclear what might happen in larger problems like ImageNet."
>
> Response: Thank you for the nice appraisal. In fact, we used the  method from Bollapragada 2018, but with a fixed batch size. The reason for this is that many other BFGS algorithms regularly failed to perform well, and in the interests of providing as fair a comparison as possible, we settled on the L-BFGS algorithm in Bollapragada 2018 with a fixed batch size.
>
> (3)
> --Reviewer Comment: "I am also not able to see the difference between the 'stochastic' line search presented here and the standard backtracking method as applied to mini-batch evaluated estimates. What is different, and new that brings in consideration for the noise? I recall that bollapragada 2018 had an additional variance based rule to check. Some more conservative values are chosen for the step length, but I do not see the justification presented in the appendix, esp Eq 47 : p is not independent from g here, being calculated as p=Hg,  so E[p^t g] is not equal to the product of the individual expectations."
>
> Response: There is a typo present in Eq 44-45, where p should be replaced by the estimate \hat{p}_z. However, we still believe the conclusion in Eq 47 is correct, by making the same note as after Eq 37: the randomness in \hat{p}_z and \hat{g}_z’ stem from different sources since \hat{p}_z must be computed before the Taylor expansion is made. Hence, the gradient estimate used to compute \hat{p}_z is not the same as the one that is used in the Taylor expansion.
>
> (4)
> --Reviewer Comment: "Some key points were left out in the discussion of the experiments. This is a common slip up when writing conf papers these days, but please do consider discussing the settings of parameters like mini-batches sizes , value of \lambda in the H derivation, how one calculates the \sigma^2_g within the algorithm presented in the Appendix. The last must include
> an extra computational cost, or are you using Adam style online variance estimator?"
>
> Response: We agree with this comment and can easily accommodate and appendix section to elaborate on these parameter value and calculations.
>
> (5)
> --Reviewer Comment: "The MSE error alone seems insufficient in the results. Please publish the test mis-classification results too. Also, why is the MSE loss used with the softmax in CIFAR? Shouldn't cross-entropy, better justified theoretically, be better justified?"
>
> Response: We again agree with this comment and we will report mis-classification results also in the revised manuscript. MSE was a typo in the CIFAR case, it should be cross-entropy.

---

### Author Response · Authors · 2018-11-23
**Response introduction**

We would like to thank all reviewers for their time and careful consideration of the paper.

Two reviewers largely agree that the manuscript proposes an interesting solution to an important problem. In contrast, a third reviewer has been very negative in their comments in areas where the the others have been more positive.

Below we address critical reviewer concerns and treat all remaining comments as positive.

---

### Meta-Review · Area_Chair1 · 2018-12-17
**Interesting idea, but the paper requires further refinement**

**Confidence:** 5
**Recommendation:** Reject

**Metareview:**

The paper investigates a novel formulation of a stochastic, quasi-Newton optimization strategy based on the natural idea of relaxing the secant conditions.  This is an interesting and promising idea, but unfortunately none of the reviewers recommended acceptance.  The reviewers unanimously fixated on weaknesses in the paper's technical presentation.  In particular, the reviewers expressed some dissatisfaction with many aspects, including:

- Key details of the experimental evaluation were omitted (particularly concerning configuration of the baseline competitors), which is an essential aspect of reproducibility.  One consequence is that the reviewers were not confident in the veracity of the experimental comparison.

- The reviewers struggled with a lack of clarity and accurate rendering of some key technical details.  An example is dissatisfaction with the non-symmetry of the inverse Hessian approximation, which was not fully alleviated by the author responses.

- The proposed approach does not appear to possess any intrinsic advantage over standard methods from a computational complexity perspective.

I think this is promising work, but a careful revision that strengthened the underlying technical claims appears necessary to make this a solid contribution.